# A Model for Learned Bloom Filters, and Optimizing by Sandwiching

**Michael Mitzenmacher**
School of Engineering and Applied Sciences
Harvard University
michaelm@eecs.harvard.edu

## Abstract

Recent work has suggested enhancing Bloom filters by using a pre-filter, based on applying machine learning to determine a function that models the data set the Bloom filter is meant to represent. Here we model such *learned Bloom filters*, with the following outcomes: (1) we clarify what guarantees can and cannot be associated with such a structure; (2) we show how to estimate what size the learning function must obtain in order to obtain improved performance; (3) we provide a simple method, sandwiching, for optimizing learned Bloom filters; and (4) we propose a design and analysis approach for a learned Bloomier filter, based on our modeling approach.

## 1 Introduction

An interesting recent paper, "The Case for Learned Index Structures" [7], argues that standard index structures and related structures, such as Bloom filters, could be improved by using machine learning to develop what the authors dub learned index structures. However, this paper did not provide a suitable mathematical model for judging the performance of such structures. Here we aim to provide a more formal model for their variation of a Bloom filter, which they call a *learned Bloom filter*.

To describe our results, we first somewhat informally describe the learned Bloom filter. Like a standard Bloom filter, it provides a compressed representation of a set of keys $\mathcal{K}$ that allows membership queries. (We may sometimes also refer to the keys as elements.) Given a key $y$, a learned Bloom filter always returns yes if $y$ is in $\mathcal{K}$, so there will be no false negatives, and generally returns no if $y$ is not in $\mathcal{K}$, but may provide false positives. What makes a learned Bloom filter interesting is that it uses a function that can be obtained by "learning" the set $\mathcal{K}$ to help determine the appropriate answer; the function acts as a pre-filter that provides a probabilistic estimate that a query key $y$ is in $\mathcal{K}$. This learned function can be used to make an initial decision as to whether the key is in $\mathcal{K}$, and a smaller backup Bloom filter is used to prevent any false negatives.

Our more formal model provides interesting insights into learned Bloom filters, and how they might be effective. In particular, here we: (1) clarify what guarantees can and cannot be associated with such a structure; (2) show how to estimate what size the learning function must obtain in order to obtain improved performance; (3) provide a simple method for optimizing learned Bloom filters; and (4) demonstrate our approach may be useful for other similar structures.

We briefly summarize the outcomes above. First, we explain how the types of guarantees offered by learned Bloom filters differ significantly from those of standard Bloom filters. We thereby clarify what application-level assumptions are required for a learned Bloom filter to be effective. Second, we provide formulae for modeling the false positive rate for a learned Bloom filter, allowing for an estimate of how small the learned function needs to be in order to be effective. We then find, perhaps surprisingly, that a better structure uses a Bloom filter before as well as after the learned function.

Because we optimize for two layers of Bloom filters surrounding the learned function, we refer to this as a *sandwiched learned Bloom filter*. We show mathematically and intuitively why sandwiching improves performance. We also discuss an approach to designing learned Bloomier filters, where a Bloomier filter returns a value associated with a set element (instead of just returning whether the element is in the set), and show it can be modeled similarly.

While the contents of this paper may be seen as relatively simple, we feel it is important to provide solid foundations in order for a wide community to understand the potential and pitfalls of data structures using machine learning components. We therefore remark that the simplicity is purposeful, and suggest it is desirable in this context. Finally, we note that this work incorporates and extends analysis that appeared in two prior working notes [8, 9].

## 2 Review: Bloom Filters

We start by reviewing standard Bloom filters and variants, following the framework provided by the reference [2].

### 2.1 Definition of the Data Structure

A Bloom filter for representing a set $S = \{x_1, x_2, \ldots, x_n\}$ of $n$ elements corresponds to an array of $m$ bits, and uses $k$ independent hash functions $h_1, \ldots, h_k$ with range $\{0, \ldots, m-1\}$. Here we follow the typical assumption that these hash functions are perfect; that is, each hash function maps each item in the universe independently and uniformly to a number in $\{0, \ldots, m-1\}$. Initially all array bits are 0. For each element $x \in S$, the array bits $h_i(x)$ are set to 1 for $1 \leq i \leq k$; it does not matter if some bit is set to 1 multiple times. To check if an item $y$ is in $S$, we check whether all $h_i(y)$ are set to 1. If not, then clearly $y$ is not a member of $S$. If all $h_i(y)$ are set to 1, we conclude that $y$ is in $S$, although this may be a *false positive*. A Bloom filter does not produce false negatives.

The primary standard theoretical guarantee associated with a Bloom filter is the following. Let $y$ be an element of the universe such that $y \notin S$, where $y$ is chosen independently of the hash functions used to create the filter. Let $\rho$ be the fraction of bits set to 1 after the elements are hashed. Then

$$\Pr(y \text{ yields a false positive}) = \rho^k.$$

For a bit in the Bloom filter to be 0, it has to not be the outcome of the $kn$ hash values for the $n$ items. It follows that

$$\mathbf{E}[\rho] = 1 - \left(1 - \frac{1}{m}\right)^{kn} \approx 1 - \mathrm{e}^{-kn/m},$$

and that via standard techniques using concentration bounds (see, e.g., [11])

$$\Pr(|\rho - \mathbf{E}[\rho]| \geq \gamma) \leq \mathrm{e}^{-\Theta(\gamma^2 m)}$$

in the typical regime where $m/n$ and $k$ are constant. That is, $\rho$ is, with high probability, very close to its easily calculable expectation, and thus we know (up to very small random deviations) what the probability is that an element $y$ will be a false positive. Because of this tight concentration around the expectation, it is usual to talk about the *false positive probability* of a Bloom filter; in particular, it is generally referred to as though it is a constant depending on the filter parameters, even though it is a random variable, because it is tightly concentrated around its expectation.

Moreover, given a set of distinct query elements $Q = \{y_1, y_2, \ldots, y_q\}$ with $Q \cap S = \emptyset$ chosen a priori before the Bloom filter is instantiated, the fraction of false positives over these queries will similarly be concentrated near $\rho^k$. Hence we may talk about the *false positive rate* of a Bloom filter over queries, which (when the query elements are distinct) is essentially the same as the false positive probability. (When the query elements are not distinct, the false positive rate may vary significantly, depending on on the distribution of the number of appearances of elements and which ones yield false positives; we focus on the distinct item setting here.) In particular, the false positive rate is a priori the same for *any* possible query set $Q$. Hence one approach to finding the false positive rate of a Bloom filter empirically is simply to test a random set of query elements (that does not intersect $S$) and find the fraction of false positives. Indeed, it does not matter what set $Q$ is chosen, as long as it is chosen independently of the hash functions.

We emphasize that, as we discuss further below, the term false positive rate often has a different meaning in the context of learning theory applications. Care must therefore be taken in understanding how the term is being used.

## 2.2 Additional Bloom Filter Benefits and Limitations

For completeness, we relate some of the other benefits and limitations of Bloom filters. More details can be found in [2].

We have assumed in the above analysis that the hash functions are fully random. As fully random hash functions are not practically implementable, there are often questions relating to how well the idealization above matches the real world for specific hash functions. In practice, however, the model of fully random hash functions appears reasonable in many cases; see [5] for further discussion on this point.

If an adversary has access to the hash functions used, or to the final Bloom filter, it can find elements that lead to false positives. One must therefore find other structures for adversarial situations. A theoretical framework for such settings is developed in [12]. Variations of Bloom filters, which adapt to false positives and prevent them in the future, are described in [1, 10]; while not meant for adversarial situations, they prevent repeated false positives with the same element.

One of the key advantages of a standard Bloom filter is that it is easy to insert an element (possibly slightly changing the false positive probability), although one cannot delete an element without using a more complex structure, such as a counting Bloom filter. However, there are more recent alternatives to the standard Bloom filter, such as the cuckoo filter [6], which can achieve the same or better space performance as a standard Bloom filter while allowing insertions and deletions. If the Bloom filter does not need to insert or delete elements, a well-known alternative is to develop a perfect hash function for the data set, and store a fingerprint of each element in each corresponding hash location (see, e.g., [2] for further explanation); this approach reduces the space required by approximately 30%.

# 3 Learned Bloom Filters

## 3.1 Definition of the Data Structure

We now consider the learned Bloom filter construction as described in [7]. We are given a set of positive keys $\mathcal{K}$ that correspond to set to be held in the Bloom filter – that is, $\mathcal{K}$ corresponds to the set $S$ in the previous section. We are also given a set of negative keys $\mathcal{U}$ for training. We then train a neural network with $\mathcal{D} = \{(x_i, y_i = 1) \mid x_i \in \mathcal{K}\} \cup \{(x_i, y_i = 0) \mid x_i \in \mathcal{U}\}$; that is, they suggest using a neural network on this binary classification task to produce a probability, based on minimizing the log loss function

$$L = \sum_{(x,y) \in \mathcal{D}} y \log f(x) + (1 - y) \log(1 - f(x)),$$

where $f$ is the learned model from the neural network. Then $f(x)$ can be interpreted as a "probability" estimate that $x$ is a key from the set. Their suggested approach is to choose a threshold $\tau$ so that if $f(x) \geq \tau$ then the algorithm returns that $x$ is in the set, and no otherwise. Since such a process may provide false negatives for some keys in $\mathcal{K}$ that have $f(x) < \tau$, a secondary structure – such as a smaller standard Bloom filter holding the keys from $\mathcal{K}$ that have $f(x) < \tau$ – can be used to check keys with $f(x) < \tau$ to ensure there are no false negatives, matching this feature of the standard Bloom filter.

In essence, [7] suggests using a pre-filter ahead of the Bloom filter, where the pre-filter comes from a neural network and estimates the probability a key is in the set, allowing the use of a smaller Bloom filter than if one just used a Bloom filter alone. Performance improves if the size to represent the learned function $f$ and the size of the smaller backup filter for false negatives is smaller than the size of a corresponding Bloom filter with the same false positive rate. Of course the pre-filter here need not come from a neural network; any approach that would estimate the probability an input key is in the set could be used.

This motivates the following formal definition:

**Definition 1** *A* learned Bloom filter *on a set of positive keys* $\mathcal{K}$ *and negative keys* $\mathcal{U}$ *is a function* $f : U \rightarrow [0, 1]$ *and threshold* $\tau$, *where $U$ is the universe of possible query keys, and an associated standard Bloom filter $B$, referred to as a backup filter. The backup filter holds the set of keys* $\{z : z \in \mathcal{K}, f(z) < \tau\}$. *For a query $y$, the learned Bloom filter returns that $y \in \mathcal{K}$ if $f(y) \geq \tau$, or if $f(y) < \tau$ and the backup filter returns that $y \in \mathcal{K}$. The learned Bloom filter returns $y \notin \mathcal{K}$ otherwise.*

## 3.2 Defining the False Positive Probability

The question remains how to determine or derive the false positive probability for a learned Bloom filter, and how to choose an appropriate threshold $\tau$. The approach in [7] is to find the false positive rate over a test set. This approach is, as we have discussed, suitable for a standard Bloom filter, where the false positive rate is guaranteed to be close to its expected value for any test set, with high probability. However, this methodology requires additional assumptions in the learned Bloom filter setting.

As an example, suppose the universe of elements is the range $[0, 1000000)$, and the set $\mathcal{K}$ of keys to store in our Bloom filter consists of a random subset of 500 elements from the range $[1000, 2000]$, and of 500 other random elements from outside this range. Our learning algorithm might determine that a suitable function $f$ yields that $f(y)$ is large (say $f(y) \approx 1/2$) for elements in the range $[1000, 2000]$ and close to zero elsewhere, and then a suitable threshold might be $\tau = 0.4$. The resulting false positive rate depends substantially on what elements are queried. If $\mathcal{Q}$ consists of elements primarily in the range $[1000, 2000]$, the false positive rate will be quite high, while if $\mathcal{Q}$ is chosen uniformly at random over the whole range, the false positive rate will be quite low. Unlike a standard Bloom filter, the false positive rate is highly dependent on the query set, and is not well-defined independently of the queries.

Indeed, it seems plausible that in many situations, the query set $\mathcal{Q}$ might indeed be similar to the set of keys $\mathcal{K}$, so that $f(y)$ for $y \in \mathcal{Q}$ might often be above naturally chosen thresholds. For example, in security settings, one might expect that queries for objects under consideration (URLs, network flow features) would be similar to the set of keys stored in the filter. Unlike in the setting of a standard Bloom filter, the false positive probability for a query $y$ can depend on $y$, even before the function $f$ is instantiated.

It is worth noting, however, that the problem we point out here can possibly be a positive feature in other settings; it might be that the false positive rate is remarkably low if the query set is suitable. Again, one can consider the range example above where queries are uniform over the entire space; the query set is very unlikely to hit the range where the learned function $f$ yields an above threshold value in that setting for a key outside of $\mathcal{K}$. The data-dependent nature of the learned Bloom filter may allow it to circumvent lower bounds for standard Bloom filter structures.

While the false positive probability for learned Bloom filters does not have the same properties as for a standard Bloom filter, we can define the false positive rate of a learned Bloom filter with respect to a given query distribution.

**Definition 2** *A false positive rate on a query distribution $\mathcal{D}$ over $\mathcal{U} - \mathcal{K}$ for a learned Bloom filter $(f, \tau, B)$ is given by*

$$\Pr_{y \sim \mathcal{D}}(f(y) \geq \tau) + (1 - \Pr_{y \sim \mathcal{D}}(f(y) \geq \tau))F(B),$$

*where $F(B)$ is the false positive rate of the backup filter $B$.*

While technically $F(B)$ is itself a random variable, the false positive rate is well concentrated around its expectations, which depends only on the size of the filter $|B|$ and the number of false negatives from $\mathcal{K}$ that must be stored in the filter, which depends on $f$. Hence where the meaning is clear we may consider the false positive rate for a learned Bloom filter with function $f$ and threshold $\tau$ to be

$$\Pr_{y \sim \mathcal{D}}(f(y) \geq \tau) + (1 - \Pr_{y \sim \mathcal{D}}(f(y) \geq \tau))\mathbf{E}[F(B)],$$

where the expectation $\mathbf{E}[F(B)]$ is meant to over instantiations of the Bloom filter with given size $|B|$.

Given sufficient data, we can determine an *empirical false positive rate* on a test set, and use that to predict future behavior. Under the assumption that the test set has the same distribution as future

queries, standard Chernoff bounds provide that the empirical false positive rate will be close to the false positive rate on future queries, as both will be concentrated around the expectation. In many learning theory settings, this empirical false positive rate appears to be referred to as simply the false positive rate; we emphasize that false positive rate, as we have explained above, typically means something different in the Bloom filter literature.

**Definition 3** *The* empirical false positive rate *on a set $\mathcal{T}$, where $\mathcal{T} \cap \mathcal{K} = \emptyset$, for a learned Bloom filter $(f, \tau, B)$ is the number of false positives from $\mathcal{T}$ divided by $|\mathcal{T}|$.*

**Theorem 4** *Consider a learned Bloom filter $(f, \tau, B)$, a test set $\mathcal{T}$, and a query set $\mathcal{Q}$, where $\mathcal{T}$ and $\mathcal{Q}$ are both determined from samples according to a distribution $\mathcal{D}$. Let $X$ be the empirical false positive rate on $\mathcal{T}$, and $Y$ be the empirical false positive rate on $\mathcal{Q}$. Then*

$$\Pr(|X - Y| \geq \epsilon) \leq e^{-\Omega(\epsilon^2 \min(|\mathcal{T}|, |\mathcal{Q}|))}.$$

**Proof:** Let $\alpha = \Pr_{y \sim \mathcal{D}}(f(y) \geq \tau)$, and $\beta$ be false positive rate for the backup filter. We first show that for $\mathcal{T}$ and $X$ that

$$\Pr(|X - (\alpha + (1 - \alpha)\beta)| \geq \epsilon) \leq 2e^{-2\epsilon^2 |\mathcal{T}|}.$$

This follows from a direct Chernoff bound (e.g., [11][Exercise 4.13]), since each sample chosen according to $\mathcal{D}$ is a false positive with probability $\alpha + (1 - \alpha)\beta$. A similar bound holds for $\mathcal{Q}$ and $Y$.

We can therefore conclude

$$\begin{aligned}
\Pr(|X - Y| \geq \epsilon) &\leq \Pr(|X - (\alpha + (1 - \alpha)\beta)| \geq \epsilon/2) \\
&\quad + \Pr(|Y - (\alpha + (1 - \alpha)\beta)| \geq \epsilon/2) \\
&\leq 2e^{-\epsilon^2 |\mathcal{T}|/2} + 2e^{-\epsilon^2 |\mathcal{Q}|/2},
\end{aligned}$$

giving the desired result. ■

Theorem 4 also informs us that it is reasonable to find a suitable parameter $\tau$, given $f$, by trying a suitable finite discrete set of values for $\tau$, and choosing the best size-accuracy tradeoff for the application. By a union bound, all choices of $\tau$ will perform close to their expectation with high probability.

While Theorem 4 requires the test set and query set to come from the same distribution $\mathcal{D}$, the negative examples $\mathcal{U}$ do not have to come from that distribution. Of course, if negative examples $\mathcal{U}$ are drawn from $\mathcal{D}$, it may yield a better learning outcome $f$.

If the test set and query set distribution do not match, because for example the types of queries change after the original gathering of test data $\mathcal{T}$, Theorem 4 offers limited guidance. Suppose $\mathcal{T}$ is derived from samples from distribution $\mathcal{D}$ and $\mathcal{Q}$ from another distribution $\mathcal{D}'$. If the two distributions are close (say in $L_1$ distance), or, more specifically, if the changes do not significantly change the probability that a query $y$ has $f(y) \geq \tau$, then the empirical false positive rate on the test set may still be relatively accurate. However, in practice it may be hard to provide such guarantees on the nature of future queries. This explains our previous statement that learned Bloom filters appear most useful when the query stream can be modeled as coming from a fixed distribution, which can be sampled during the construction.

We can return to our previous example to understand these effects. Recall our set consists of 500 random elements from the range $[1000, 2000]$ and 500 other random elements from the range $[0, 1000000)$. Our learned Bloom filter has $f(y) \geq \tau$ for all $y$ in $[1000, 2000]$ and $f(y) < \tau$ otherwise. Our backup filter will therefore store 500 elements. If our test set is uniform over $[0, 1000000)$ (excluding elements stored in the Bloom filter), our false positive rate from elements with too large an $f$ value would be approximately 0.0002; one could choose a backup filter with roughly the same false positive probability for a total empirical false positive probability of 0.0004. If, however, our queries are uniform over a restricted range $[0, 100000)$, then the false positive probability would jump to 0.0022 for the learned Bloom filter, because the learned function would yield more false positives over the smaller query range.

### 3.3 Additional Learned Bloom Filter Benefits and Limitations

Learned Bloom filters can easily handle insertions into $\mathcal{K}$ by adding the key, if is does not already yield a (false) positive, to the backup filter. Such changes have a larger effect on the false positive probability than for a standard Bloom filter, since the backup filter is smaller. Keys cannot be deleted naturally from a learned Bloom filter. A deleted key would simply become a false positive, which (if needed) could possibly be handled by an additional structure.

As noted in [7], it may be possible to re-learn a new function $f$ if the data set changes substantially via insertions and deletion of keys from $\mathcal{K}$. Of course, besides the time needed to re-learn a new function $f$, this requires storing the original set somewhere, which may not be necessary for alternative schemes. Similarly, if the false positive probability proves higher than desired, one can re-learn a new function $f$; again, doing so will require access to $\mathcal{K}$, and maintaining a (larger) set $\mathcal{U}$ of negative examples.

## 4 Size of the Learned Function

We now consider how to model the performance of the learned Bloom filter with the goal of understanding how small the representation of the function $f$ needs needs to be in order for the learned Bloom filter to be more effective than a standard Bloom filter. [1]

Our model is as follows. The function $f$ associated with Definition 1 we treat as an *oracle* for the keys $\mathcal{K}$, where $|\mathcal{K}| = m$, that works as follows. For keys not in $\mathcal{K}$ there is an associated false positive probability $F_p$, and there are $F_n m$ false negatives for keys in $\mathcal{K}$. (The value $F_n$ is like a false negative probability, but given $\mathcal{K}$ this fraction is determined and known according to the oracle outcomes.) We note the oracle representing the function $f$ is meant to be general, so it could potentially represent other sorts of filter structures as well. As we have described in Section 3.2, in the context of a learned Bloom filter the false positive rate is necessarily tied to the query stream, and is therefore generally an empirically determined quantity, but we take the value $F_p$ here as a given. Here we show how to optimize over a single oracle, although in practice we may possibly choose from oracles with different values $F_p$ and $F_n$, in which case we can optimize for each pair of values and choose the best suited to the application.

We assume a total budget of $bm$ bits for the backup filter, and $|f| = \zeta$ bits for the learned function. If $|\mathcal{K}| = m$, the backup Bloom filter only needs to hold $mF_n$ keys, and hence we take the number of bits per stored key to be $b/F_n$. To model the false positive rate of a Bloom filter that uses $j$ bits per stored key, we assume the false positive rate falls as $\alpha^j$. This is the case for a standard Bloom filter (where $\alpha \approx 0.6185$ when using the optimal number of hash functions, as described in the survey [2]), as well as for a static Bloom filter built using a perfect hash function (where $\alpha = 1/2$, again described in [2]). The analysis can be modified to handle other functions for false positives in terms of $j$ in a straightforward manner. (For example, for a cuckoo filter [6], a good approximation for the false positive rate is $c\alpha^j$ for suitable constants $c$ and $\alpha$.)

The false positive rate of a learned Bloom filter is $F_p + (1 - F_p)\alpha^{b/F_n}$. This is because, for $y \notin \mathcal{K}$, $y$ causes a false positive from the learned function $f$ with probability $F_p$, or with remaining probability $(1 - F_p)$ it yields a false positive on the backup Bloom filter with probability $\alpha^{b/F_n}$.

A comparable Bloom filter using the same number of total bits, namely $bm + \zeta$ bits, would have a false positive probability of $\alpha^{b+\zeta/m}$. Thus we find an improvement using a learned Bloom filter whenever

$$F_p + (1 - F_p)\alpha^{b/F_n} \leq \alpha^{b+\zeta/m},$$

which simplifies to

$$\zeta/m \leq \log_\alpha \left( F_p + (1 - F_p)\alpha^{b/F_n} \right) - b,$$

where we have expressed the requirement in terms of a bound on $\zeta/m$, the number of bits per key the function $f$ is allowed.

This expression is somewhat unwieldy, but it provides some insight into what sort of compression is required for the learned function $f$, and how a practitioner can determine what is needed. First, one can determine possible thresholds and the corresponding rate of false positive and false negatives from the learned function. For example, the paper [7] considers situations where $F_p \approx 0.01$, and $F_n \approx 0.5$; let us consider $F_p = 0.01$ and $F_n = 0.5$ for clarity. If we have a target goal of one byte per item, a standard Bloom filter achieves a false positive probability of approximately 0.0214. If our learned function uses 3 bits per item (or less), then the learned Bloom filter can use $5m$ bits for the backup Bloom filter, and achieve a false positive rate of approximately 0.0181. The learned Bloom filter will therefore provide over a $10\%$ reduction in false positives with the same or less space. More generally, in practice one could determine or estimate different $F_p$ and $F_n$ values for different thresholds and different learned functions of various sizes, and use these equations to determine if better performance can be expected without in depth experiments.

Indeed, an interesting question raised by this analysis is how learned functions scale in terms of typical data sets. In extreme situations, such as when the set $\mathcal{K}$ being considered is a range of consecutive integers, it can be represented by just two integers, which does not grow with $\mathcal{K}$. If, in practice, as data sets grow larger the amount of information needed for a learned function $f$ to approximate key sets $\mathcal{K}$ grows sublinearly with $|\mathcal{K}|$, learned Bloom filters may prove very effective.

## 5   Sandwiched Learned Bloom Filters

### 5.1   The Sandwich Structure

Given the formalization of the learned Bloom filter, it seems natural to ask whether this structure can be improved. Here we show that a better structure is to use a Bloom filter before using the function $f$, in order to remove most queries for keys not in $\mathcal{K}$. We emphasize that this *initial Bloom filter* does not declare that an input $y$ is in $\mathcal{K}$, but passes forward all matching keys to the learned function $f$, and it returns $y \notin \mathcal{K}$ when the Bloom filter shows the key is not in $\mathcal{K}$. Then, as before, we use the function $f$ to attempt to remove false positives from the initial Bloom filter, and then use the backup filter to allow back in keys from $\mathcal{K}$ that were false negatives for $f$. Because we have two layers of Bloom filters surrounding the learned function $f$, we refer to this as a *sandwiched learned Bloom filter*. The sandwiched learned Bloom filter is represented pictorially in Figure 1.

In hindsight, our result that sandwiching improves performance makes sense. The purpose of the backup Bloom filter is to remove the false negatives arising from the learned function. If we can arrange to remove more false positives up front, then the backup Bloom filter can be quite porous, allowing most everything that reaches it through, and therefore can be quite small. Indeed, surprisingly, our analysis shows that the backup filter should not grow beyond a fixed size.

### 5.2   Analyzing Sandwiched Learned Bloom Filters

We model the sandwiched learned Bloom filter as follows. As before, the learned function $f$ in the middle of the sandwich we treat as an *oracle* for the keys $\mathcal{K}$, where $|\mathcal{K}| = m$. Also as before, for keys not in $\mathcal{K}$ there is an associated false positive probability $F_p$, and there are $F_n m$ false negatives for keys in $\mathcal{K}$.

We here assume a total budget of $bm$ bits to be divided between an initial Bloom filter of $b_1 m$ bits and a backup Bloom filter of $b_2 m$ bits. As before, we model the false positive rate of a Bloom filter that uses $j$ bits per stored key as $\alpha^j$ for simplicity. The backup Bloom filter only needs to hold $mF_n$ keys, and hence we take the number of bits per stored key to be $b_2 / F_n$. If we find the best value of $b_2$ is $b$, then no initial Bloom filter is needed, but otherwise, an initial Bloom filter is helpful.

The false positive rate of a sandwiched learned Bloom filter is then $\alpha^{b_1}(F_p + (1 - F_p)\alpha^{b_2/F_n})$. To see this, note that for $y \notin \mathcal{K}$, $y$ first has to pass through the initial Bloom filter, which occurs with probability $\alpha^{b_1}$. Then $y$ either causes a false positive from the learned function $f$ with probability $F_p$, or with remaining probability $(1 - F_p)$ it yields a false positive on the backup Bloom filter, with probability $\alpha^{b_2/F_n}$.

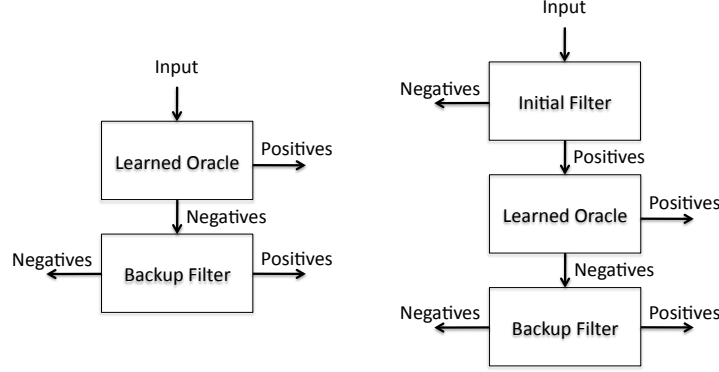

Figure 1: The left side shows the original learned Bloom filter. The right side shows the sandwiched learned Bloom filter.

As $\alpha, F_p, F_n$ and $b$ are all constants for the purpose of this analysis, we may optimize for $b_1$ in the equivalent expression

$$F_p \alpha^{b_1} + (1 - F_p)\alpha^{b/F_n}\alpha^{b_1(1-1/F_n)}.$$

The derivative with respect to $b_1$ is

$$F_p(\ln \alpha)\alpha^{b_1} + (1 - F_p)\left(1 - \frac{1}{F_n}\right)\alpha^{b/F_n}(\ln \alpha)\alpha^{b_1(1-1/F_n)}.$$

This equals 0 when

$$\frac{F_p}{(1 - F_p)\left(\frac{1}{F_n} - 1\right)} = \alpha^{(b-b_1)/F_n} = \alpha^{b_2/F_n}. \tag{1}$$

This further yields that the false positive rate is minimized when $b_2 = b_2^*$, where

$$b_2^* = F_n \log_\alpha \frac{F_p}{(1 - F_p)\left(\frac{1}{F_n} - 1\right)}. \tag{2}$$

This result may be somewhat surprising, as here we see that the optimal value $b_2^*$ is a constant, independent of $b$. That is, the number of bits used for the backup filter is not a constant fraction of the total budgeted number of bits $bm$, but a fixed number of bits; if the number of budgeted bits increases, one should simply increase the size of the initial Bloom filter as long as the backup filter is appropriately sized.

In hindsight, returning to the expression for the false positive rate $\alpha^{b_1}(F_p + (1 - F_p)\alpha^{b_2/F_n})$ provides useful intuition. If we think of sequentially distributing the $bm$ bits among the two Bloom filters, the expression shows that bits assigned to the initial filter (the $b_1$ bits) reduce false positives arising from the learned function (the $F_p$ term) as well as false positives arising subsequent to the learned function (the $(1 - F_p)$ term), while the backup filter only reduces false positives arising subsequent to the learned function. Initially we would provide bits to the backup filter to reduce the $(1 - F_p)$ rate of false positives subsequent to the learned function. Indeed, bits in the backup filter drive down this $(1 - F_p)$ term rapidly, because the backup filter holds fewer keys from the original set, leading to the $b_2/F_n$ (instead of just a $b_2$) in the exponent in the expression $\alpha^{b_2/F_n}$. Once the false positives coming through the backup Bloom filter reaches an appropriate level, which, by plugging in the determined optimal value for $b_2$, we find is $F_p/\left(\frac{1}{F_n} - 1\right)$, then the tradeoff changes. At that point the gains from reducing the false positives by increasing the bits for the backup Bloom filter become smaller than the gains obtained by increasing the bits for the initial Bloom filter.

Again, we can look at situations discussed in [7] for some insight. Suppose we have a learned function $f$ where $F_n = 0.5$ and $F_p = 0.01$. We consider $\alpha = 0.6185$ (which corresponds to a standard Bloom filter). We do not consider the size of $f$ in the calculation below. Then the optimal value for $b_2$ is

$$b_2^* = (\log_\alpha 1/99)/2 \approx 6.$$

Depending on our Bloom filter budget parameter $b$, we obtain different levels of performance improvement by using the initial Bloom filter. At $b = 8$ bits per key, the false positive rate drops from approximately 0.010045 to 0.005012, over a factor of 2. At $b = 10$ bits per key, the false positive rate drops from approximately 0.010066 to 0.001917, almost an order of magnitude.

We may also consider the implications for the oracle size. Again, if we let $\zeta$ represent the size of the oracle in bits, then a corresponding Bloom filter would have a false positive probability of $\alpha^{b+\zeta/m}$. Hence we have an improvement whenever

$$\alpha^{b_1}(F_p + (1 - F_p)\alpha^{b_2/F_n}) \leq \alpha^{b+\zeta/m}.$$

For $b$ sufficiently large that $b_1 > 0$, we can calculate the false positive probability of the optimized sandwiched Bloom filter. Let $b_2^*$ be the optimal value for $b_2$ from equation 2 and $b_1^*$ be the corresponding value for $b_1$. First using the relationship from equation 1, we have a gain whenever

$$\alpha^{b_1^*} \frac{F_p}{1 - F_n} \leq \alpha^{b+\zeta/m}.$$

Using $b_1^* = b - b_2^*$ and equation 2 gives

$$\zeta/m \leq \log_\alpha \frac{F_p}{1 - F_n} - F_n \log_\alpha \frac{F_p}{(1 - F_p)\left(\frac{1}{F_n} - 1\right)}.$$

Again, this expression is somewhat unwieldy, but one useful difference from the analysis of the original learned Bloom filter is that we see the improvement does not depend on the exact value of $b$ (as long $b$ is large enough so that $b_1 > 0$, and we use the optimal value for $b_2$). For $F_p = 0.01$, $F_n = 0.5$, and $\alpha = 0.6185$, we find a gain whenever $\zeta/m$ falls below approximately 3.36.

A possible further advantage of the sandwich approach is that it makes learned Bloom filters more robust. As discussed previously, if the queries given to a learned Bloom filter do not come from the same distribution as the queries from the test set used to estimate the learned Bloom filter's false positive probability, the actual false positive probability may be substantially larger than expected. The use of an initial Bloom filter mitigates this problem, as this issue then only affects the smaller number of keys that pass the initial Bloom filter.

We note that a potential disadvantage of the sandwich approach may be that it is more computationally complex than a learned Bloom filter without sandwiching, requiring possibly more hashing and memory accesses for the initial Bloom filter. The overall efficiency would be implementation dependent, but this remains a possible issue for further research.

## 6  Learned Bloomier Filters

In the supplemental material, we consider *learned Bloomier filters*. Bloomier filters are a variation of the Bloom filter idea where each key in the set $\mathcal{K}$ has an associated value. The Bloomier filter returns the value for every key of $\mathcal{K}$, and is supposed to return a null value for keys not in $\mathcal{K}$, but in this context there can be false positives where the return for a key outside of $\mathcal{K}$ is a non-null value with some probability. We derive related formulae for the performance of learned Bloomier filters.

## 7  Conclusion

We have focused on providing a more formal analysis of the proposed learned Bloom filter. As part of this, we have attempted to clarify a particular issue in the Bloom filter setting, namely the dependence of what is referred to as the false positive rate in [7] on the query set, and how it might affect the applications this approach is suited for. We have also found that our modeling laeds to a natural and interesting optimization, based on sandwiching, and allows for generalizations to related structures, such as Bloomier filters. Our discussion is meant to encourage users to take care to realize all of the implications of the learned Bloom filter approach before adopting it. However, for sets that can be accurately predicted by small learned functions, the learned Bloom filter may provide a novel means of obtaining significant performance improvements over standard Bloom filter variants.

## Acknowledgments

The author thanks Suresh Venkatasubramanian for suggesting a closer look at [7], and thanks the authors of [7] for helpful discussions involving their work. This work was supported in part by NSF grants CCF-1563710, CCF-1535795, CCF-1320231, and CNS-1228598. Part of this work was done while visiting Microsoft Research New England.

## Footnotes

[1]We thank Alex Beutel for pointing out that our analysis in [9] could be used in this manner.

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
