[Supplementary Material · neurips_final_supplementary.pdf]



Figure 2: A learned Bloomier filter design. Keys obtain a value, which may be null, from the learned oracle. If a key does *not* hit on a Bloom filter, the key returns with the value from the oracle; in this way, false positives from the oracle may result. The Bloom filter stores false negatives from the learned oracle, and passes them to the backup Bloomier filter to obtain the correct value. False positive hits at the Bloom filter both require the backup Bloomier filter to hold values for additional keys from $\mathcal{K}$, and may yield false positives for keys outside of $\mathcal{K}$ at the backup Bloomier filter.

# 8   Supplemental: Learned Bloomier Filters Derivation

Bloomier filters are a variation of the Bloom filter idea where each key in the set $\mathcal{K}$ has an associated value, which for convenience we will assume is a $u$-bit value. The Bloomier filter returns the value for every key of $\mathcal{K}$, as is supposed to return a null value for keys not in $\mathcal{K}$, but in this context there can be false positives where the return for a key outside of $\mathcal{K}$ is a non-null value with some probability. For our purposes, this description will suffice (although we present a few more details below in presenting our model), but more information on Bloomier filters and their constructions can be found in [3, 4].

Here we imagine that we can derive a learned function $f$ that will return a value given an input, with the goal being that the function will return the appropriate $u$-bit value for a key in $\mathcal{K}$ and the null value otherwise. In this setting we refer to a false positive as a key outside of $\mathcal{K}$ that obtains a non-null value, and a false negative as a key in $\mathcal{K}$ that obtains an incorrect value, where that value may either be the null value or the wrong value for that key.

Notice in this setting that, because keys in $\mathcal{K}$ may obtain an incorrect value that is not merely null, the system to correct for false negatives must be slightly more complicated. We propose an approach in Figure 2. The input is first passed to the learned oracle, which provides a predicted value. To handle false negatives, we provide a two-stage scheme. First, we use a Bloom filter to hold any keys that lead to false negatives. If the Bloom filter returns a key is a positive, which we refer to as a hit on the Bloom filter to avoid ambiguity, it is assumed that that key is a false negative from the oracle and a backup Bloomier filter is used to determine its value. If the Bloom filter returns a key is a negative, it is assumed the learned oracle provided the correct value (whether null or non-null) for that key, and that value is returned. We can see that for every key in $\mathcal{K}$ the correct value is returned, so the question is what is the false positive rate for this chain.

Our model is as follows. We again treat the function $f$ as an oracle for the keys in $\mathcal{K}$, where $|\mathcal{K}| = m$, and the size of the oracle is $\zeta$. For keys not in $\mathcal{K}$ there is an associated false positive probability $F_p$, and there are $F_n m$ false negatives for keys in $\mathcal{K}$. The Bloom filter will consist of $bF_n m$ bits and have its own false positive probability of $\alpha^b$. (Of course, instead of a standard Bloom filter we could use a learned Bloom filter in its place, but that is harder to model.)

To model a Bloomier filter, we use the following approach: a Bloomier filter for $z$ keys uses space $cz(u + r)$, where $u$ is the number of bits in the return value, $c$ is constant that is determined by the Bloomier filter design, and $r$ is a parameter chosen, with the result that the false positive probability for a key outside of $\mathcal{K}$ is $2^{-r}$. This setup, for example, matches the construction of [4]. One can think of it as having $cz$ cells of $u + r$ bits. The simple construction of [4] hashes a query key to (typically)

three cells, and exclusive-ors their contents together; if the result is a valid $u$-bit value (say with $r$ leading zeroes), this value is returned, and otherwise a null value is returned. The table is initially filled so that the right values are returned for the $z$ keys, and other keys obtain a value uniformly distributed over $u + r$ bits, leading to the false positive probability of $2^{-r}$. This requires $cz$ cells for some $c > 1$ to provide enough "room" to set up suitable cell values initially.

If we just used a standard Bloomier filter for the keys $\mathcal{K}$, then we would use $cm(u + r)$ bits for a false positive probability of $2^{-r}$.

For our learned Bloom filter construction, we start with the learned function of size $\zeta$. The function $f$ yields $mF_n$ false negatives; these $mF_n$ false negatives can be stored in the Bloom filter using $bmF_n$ bits and the corredsponding values recovered by the backup Bloomier filter. The keys not hit by this Bloom filter, which we refer to as the keys passed by this Bloom filter, may now include false positives for our learned Bloomier filter. A key not in $\mathcal{K}$ will yield a false positive here with probability $F_p(1 - \alpha^b)$; that is, the key must have been a false positive for the learned oracle, but must have not been a hit on the Bloom filter. Also, note that some keys from $\mathcal{K}$ that obtained the correct value from $f$ may hit the Bloom filter, and therefore will also have to have their values provided by the backup Bloomier filter. Of these $m(1 - F_n)$ keys from $\mathcal{K}$, a fraction $\alpha^b$ are expected to be false positives in the Bloom filter; as we have throughout the paper, we will use the expectation, keeping in mind the true result will be concentrated around this expectation. Hence, in total, we need the backup Bloomier filter for $m' = m(F_n + (1 - F_n)\alpha^b)$ keys. Suppose we use $cm'(u + r')$ bits for the backup Bloomier filter. Then our total space is

$$\zeta + bmF_n + cm(F_n + (1 - F_n)\alpha^b)(u + r'),$$

and our overall false positive probability is

$$F_p(1 - \alpha^b) + \alpha^b 2^{-r'},$$

where the first term is from false positives from the oracle than passed Bloom filter, and the second term is from queries that hit the Bloom filter and give false positives in the backup Bloomier filter.

Again, these expressions are somewhat unwieldy because of the number of parameters. At a high level, however, these expressions reinforce helpful intuition. The cost per element in a Bloomier filter is rather high, because the value must be stored. Therefore if the false negatives as given by $F_n$ can be driven down to reasonable value with a small enough learned function, there should be space available to pay the $\zeta$ bits of the learned function as well as the additional Bloom filter.