[Reviews · NeurIPS 2018]

Reviewer 1



I enjoyed reading this paper and thought it was very well written. The one negative about the paper is that the results presented are somewhat simplistic (the author’s acknowledge this point directly). The paper considers an interesting recent effort (specifically in the paper “The Case for Learned Index Structures”) to use predictive machine learning models to improve the performance of basic data structures. In particular, this work focuses on the standard Bloom filter for quickly detecting set membership, possibly with some false positives. “The Case for Learned Index Structures” suggests a “learned” bloom filter, which essentially uses a learning pre-filter to guess if an input query is in the set of interest. If the pre-filter says the element is not, then a backup bloom filter is queried. The hope is that, if the learned bloom filter can guess a good fraction membership queries correct, without too many false positives, than the backup filter can be more compact (i.e. will require less storage). “The Case for Learned Index Structures” observed that empirically, this strategy could improve on the space/accuracy tradeoff over a standard bloom filter. The goal of this paper is to formalize the learned bloom filter problem. In particular, it suggests the natural model where we have access to a training set that contains negative examples (i.e. not in the target set) drawn from the same distribution that future negative queries will come from. It then states some very basic results on estimating the false positive rate of a learned filter by sampling from this query distribution (specifically, just apply a Chernoff bound). Such an estimate is necessary to properly optimize how aggressively the learned pre-filter should try to predict that elements are in the target set — the more it does so, the smaller the backup bloom filter needs to be for a given false positive rate, but also the higher the false positive rate of the learned pre-filter. Next the paper discusses how compact the learned pre-filter would need to be to justify its use over simply using a larger block filter. I didn’t think this section was particularly, insightful. The size of a bloom filter roughly needs to scale linearly with m, the number of elements in the set of interest. Of course if the the learning algorithm can find a pre-filter that reduces m and uses less than m space, there’s an opportunity for improvement. Finally, the authors present a suggested improvement to the learned bloom filter. The main observation is that, for a fixed pre-filter, there is not much to gain from using more bits in the backup bloom filter once its false positive rate is about comparable to that of the pre-filter: at most we can improve on the false positive rate by a factor of 2. A more effective use of extra space is to use another bloom filter *before* the learned filter, which will more effectively reduce the overall false positive rate. I think this is a nice suggestion, although I would have liked to see some more justification, perhaps experimentally. There are many ways to use extra space: increase the size of the learned filter, decrease the threshold of the learned filter so that more queries pass though (and simultaneousy increase the size of the backup filter), run multiple separately learned filters in parallel, etc. That said, the authors idea of “sandwiched” filters does at first glance seem like a reasonably practical way of implementing learned filters in practice, since it can take essentially any learn filter with any false positive rate and combine that function with an easily computed number of additional bits to achieve a specified target accuracy. Overall, I found this paper valuable because it introduced me to the learned bloom filter problem with very clear exposition and hopefully puts this problem on firmer ground for future research. At the same time, the actual novel research contribution of the paper is relatively small. However, it is not so far below that of many NIPS papers, so for me the quality of exposition and “problem building” push the paper over the threshold for acceptance. Small comments and typos: - paragraph from lines 33-48 seems redundant with 25-32. - I thought the rather long discussion about the training set and future query set needing to come from the same (or nearly the same) distribution superfluous. I think it should be obvious to the vast majority of readers that guarantees will not carry over when this is not the case.

Reviewer 2



This paper provides a theoretical analysis of "learned Bloom filters", a recently proposed index structure which combines standard Bloom filters with a learned function to reduce the number of bits required to achieve 0 false negative and the same false positive. Paper also proposes a new index structure, "sandwiched learned Bloom filter", which additionally places an initial Bloom filter before the learned Bloom filter to further lower the false positive rate. Analysis consists of two parts: - Theorem 4 shows that whp the empirical false positive rate can be well estimated with a test (validation) set drawn from the same distribution as the query set. - Section 4 computes the compression rate (\zeta / m) that the learned function must have (for a false positive and false negative rate) so that the learned Bloom filter has lower storage than an equivalent Bloom filter. Strengths: - Paper is clear and well written and provides some theoretical foundation for a new index structure that has garnered attention in the database community. - Theorem 4 provides a sound approach to estimate empirical false positive rate; shows that the natural approach of using a test set is theoretically sound. - Sandwiched learned Bloom filters are a new contribution that extends the new research area of learned index structures. Weaknesses: - Theoretical analyses are not particularly difficult, even if they do provide some insights. That is, the analyses are what I would expect any competent grad student to be able to come up with within the context of a homework assignment. I would consider the contributions there to be worthy of a posted note / arXiv article. - Section 4 is interesting, but does not provide any actionable advice to the practitioner, unlike Theorem 4. The conclusion I took was that the learned function f needs to achieve a compression rate of \zeta / m with a false positive rate F_p and false negative rate F_n. To know if my deep neural network (for example) can do that, I would have to actually train a fixed size network and then empirically measure its errors. But if I have to do that, the current theory on standard Bloom filters would provide me with an estimate of the equivalent Bloom filter that achieves the same error false positive as the learned Bloom filter. - To reiterate the above point, the analysis of Section 4 doesn't change how I would build, evaluate, and decide on whether to use learned Bloom filters. - The analytical approach of Section 4 gets confusing by starting with a fixed f with known \zeta, F_p, F_n, and then drawing the conclusion for an a priori fixed F_p, F_n (lines 231-233) before fixing the learned function f (lines 235-237). In practice, one typically fixes the function class (e.g. parameterized neural networks with the same architecture) *first* and measures F_p, F_n after. For such settings where \zeta and b are fixed a priori, one would be advised to minimize the learned Bloom filter's overall false positive (F_p + (1-F_p)\alpha^{b/F_n}) in the function class. An interesting analysis would then be to say whether this is feasible, and how it compares to the log loss function. Experiments can then conducted to back this up. This could constitute actionable advice to practitioners. Similarly for the sandwiched learned Bloom filter. - Claim (first para of Section 3.2) that "this methodology requires significant additional assumptions" seems too extreme to me. The only additional assumption is that the test set be drawn from the same distribution as the query set, which is natural for many machine learning settings where the train, validation, test sets are typically assumed to be from the same iid distribution. (If this assumption is in fact too hard to satisfy, then Theorem 4 isn't very useful too.) - Inequality on line 310 has wrong sign; compare inequality line 227 --- base \alpha < 1. - No empirical validation. I would have like to see some experiments where the bounds are validated.

Reviewer 3



This a purely theoretical paper that is concerned with modeling learned Bloom filters (which are a combination of the statistical filter with the backup classic-Bloom eliminating false negatives arising from the use of the statistical model). In addition to analyzing this method, it proposes a sandwiched version of the learned Bloom-filter (classic-Bloom, statistical-filter, classic-Bloom). It also derives a relationship between the size of the learned model and its associated false positive and negative rates, which entail the same or better OVERALL false positive rate of the learned Bloom filter. I thank the author(s) for the interesting read. I have found it quite educational. However, I have some mixed feelings about the paper. Below, I talk about these in details. But, here are the few highlights: i) Authors focus primarily on the analysis of the false positive rate at a given bit budget. I think it is an important step. However, as noted by Kraska et al an advantage of the learned filter is that it can use the given bit budget more efficiently. In particular, computing the prediction of the neural network uses easy-to-parallelize operations with localized memory accesses and do not suffer from branch misprediction. Furthermore, if run on GPUs they can use of a much faster GPU memory (matrix multiplication on CPU can be very memory-efficient too). I think this warrants at least a discussion. ii) Perhaps, I am missing something but Section 3.2 is not very useful, because it does not tell how to choose tau without actually running experiments on the test set. iii) It looks like authors make somewhat inaccurate claims about properties of the classic Bloom filters. *PROS* Paper makes the initial and certainly important step towards analyzing learned Bloom filters. It provides useful examples, which shed light on behavior/properties of the learned Bloom filters: I find these quite educational. In particular, it illustrates how easy performance of the Bloom filters can deteriorate when there is a mismatch It proposes two extensions of the Bloom filters, in particular, a sandwiched Bloom filter. *CONS* It feels that the paper was somewhat hastily written (see some comments below). The prose is somewhat hard to understand. In part, it is because of the long sentences (sometimes not very coherent). Sometimes, it is because major findings and observations are buried somewhere. I would encourage you to have some short list of easy-to-grasp high-level overview of the findings. For example, it would be a pity to miss notes in lines 238-242. The discussion in lines 71-81 makes an assumption that our hash functions operate like random and independent variables. Of course, this is not true in practice, but how much is the difference in realistic settings? Furthermore, this makes you say that you can estimate the false positive rate from any subset of queries. My guess would be that this is true only for good-enough families for hash functions. Although classic Bloom filters should be much less susceptible to distribution changes, I don't think they are total immune to them. In particular, I believe there may be adversarial sequences of keys with much worse FP rate. Discussing these aspects seems to would have been helpful. Last, but not least, as mentioned in the highlights the efficiency of the learned Bloom filters for a given number of bits can be better compared to the classic Bloom filter. In particular, because one lookup may require up to k random memory accesses. From this practical perspective, it is not clear to me if the sandwiched approach always makes sense (as it entails additional memory accesses). In particular, for roughly the same false positive ratio and comparable bit budgets, the original learned Bloom filter may be more efficient than than sandwiched version. In particular, I would like to see theory and/or experiments on this. What if we take the budget for the prefilter in the sandwiched version and give it all to the statistical filter? *SOME DETAILED COMMENTS* 55: universe to independently and uniformly to a number : please check if this is grammatically correct. 63: how? Since you provide an overview anyways, it would be nice to explain that you consider a probability of setting each bit as a random independent variable. Then, not setting it to 1 kn times in a row has such and such probability. 64: Which bound do you use? I guess it's for the binomial with the Chernoff or Hoeffding inequality, but it's never bad to clarify. 67: what the probability is than an element y will be a false positive -> what the probability is that 67: Because of this, it is usual to talk -> because it is tightly concentrated around its expectation, it is usual to talk 109: universe possible query keys -> universe OF possible query keys 122: and 500 -> and OF 500 123: the first half of the line has extra text 120-144: a nice useful illustration 227: it is truly to great to some analysis, but what is missing there is the relationship between \theta and F_n. It doesn't have to be theoretical: some empirical evaluation can be useful too. 238-242: This are really nice points, but it is buried deep in the middle of the paper 254-258: It is not only about improving false positive ratio at a given number of bits. The pre-filtering using a Bloom filter has a non-trivial cost, which includes several random memory accesses. The advantage of having a learned filter, which is implemented by a compact neural net, is that computation over that network is very memory-local and can be easily outsourced to GPU (and kept in a super-fast GPU memory).